# What matters most? A qualitative study exploring priorities for supportive interventions for people with tuberculosis in urban Viet Nam

Isabel Smith,[1] Rachel Forse [1,2] Kristi Sidney Annerstedt,[1] Nguyen Thi Thanh,[3] Lan Nguyen,[4] Thi Hoang Yen Phan,[3] Han Nguyen,[2] Andrew Codlin,[1,2] Luan Nguyen Quang Vo [1,2] Nga Thi Thuy Nguyen,[2] Amera Khan,[5] Jacob Creswell,[5] Minh Pham Huy,[6] Lopa Basu,[6] Knut Lönnroth,[1] Binh Hoa Nguyen [7] Viet Nhung Nguyen,[7] Salla Atkins[1,8]

**Correspondence to**
Rachel Forse;
rachel.forse@tbhelp.org

## ABSTRACT

**Introduction** The health and economic burden of tuberculosis (TB) in urban Viet Nam is high. Social protection and support interventions can improve treatment outcomes and reduce costs. However, evidence regarding optimal strategies in this context is lacking. This study aimed to increase understanding of what people with TB and healthcare providers (HCPs) perceive as important to improve TB treatment outcomes and reduce costs.

**Methods** We conducted qualitative focus group discussions (seven groups, n=30) and key informant interviews (n=4) with people with drug-susceptible and multidrug-resistant TB and HCPs in Ha Noi and Ho Chi Minh City. Topic guides covered perspectives on and prioritisation of different forms of social protection and support. Data were analysed using reflexive thematic analysis and interpreted using a Framework for Transformative Social Protection.

**Results** We identified three themes and seven subthemes. The first theme, 'Existing financial safety nets are essential, but could go further to support people affected by TB', highlights that support to meet the medical costs of TB treatment and flexible cash transfers are a priority for people with TB and HCPs. The second, 'It is important to promote "physical and spiritual health" during TB treatment', demonstrates that extended psychosocial and nutritional support would encourage people with TB during their treatment. The third, 'Accessibility and acceptability are critical in designing social support interventions for people with TB', shows the importance of ensuring that support is accessible and proportional to the needs of people with TB and their families.

**Conclusions** Accessible interventions that incorporate financial risk protection, nutritional and psychosocial support matter most to people with TB and HCPs in urban Viet Nam to improve their treatment outcomes and reduce catastrophic costs. This study can inform the design of stronger person-centred interventions to advance progress towards the goals of the WHO's End TB Strategy.

## INTRODUCTION

Tuberculosis (TB) caused 1.6 million deaths in 2021.[1] It disproportionately affects poorer

## STRENGTHS AND LIMITATIONS OF THIS STUDY

⇒ This study included a diverse group of participants, enhancing the trustworthiness and validity of the findings, and ensuring that the resulting recommendations took the perspectives of both people with tuberculosis (TB) and healthcare providers into account. This is particularly important given the role of healthcare providers in delivering and connecting people with TB to available support.

⇒ Interviews were conducted in a neutral space rather than in a health facility to mitigate social desirability bias.

⇒ This study was conducted solely in urban settings in Viet Nam, therefore the findings may not be generalisable to rural settings within Viet Nam or to other high-burden countries.

⇒ The majority of the interviews were conducted during COVID-19 and it is possible that the needs of people with TB may have shifted since data collection.

⇒ The analysis was conducted on English translations of the interviews. This may have influenced the interpretation of data; however, the study took a rigorous, collaborative approach to limit potential effects of ethnocentrism on the analysis.

populations who are more vulnerable to TB infection, development of active TB disease and worse treatment outcomes.[1–4] Further, TB can push affected households into poverty[5 6] through the accrual of direct and indirect costs. These can amount to catastrophic costs, defined by the WHO as total costs due to TB exceeding 20% of annual household income.[1] The impoverishing effects of TB may have been compounded by COVID-19, particularly for more vulnerable populations.[7] Eliminating TB will require both biomedical innovation and

socioeconomic interventions, as outlined in the WHO's End TB Strategy.[8]

The provision of social protection schemes must be a part of the response to end TB. There is consistent evidence that spending on broad social protection, including population-level poverty reduction schemes such as pensions and child support, is inversely associated with TB incidence and mortality.[9] Some estimates indicate that expansion of social protection initiatives could reduce global TB incidence by up to 76%.[10] Furthermore, social protection schemes for people with TB such as cash transfers, transport vouchers and nutritional support have been shown to defray out-of-pocket costs, improve treatment outcomes or improve treatment adherence in various high-burden settings.[11–13] For example, cash transfers for people with TB in Peru led to improved treatment success and decreased likelihood of incurring catastrophic costs.[14 15] While most social protection interventions for TB-affected household focus on financial or nutritional support,[16] social and psychological support interventions have also contributed to improved treatment outcomes and reduced costs.[14 17]

Viet Nam is one of the world's highest burden countries for TB and multidrug-resistant TB (MDR-TB),[18] and TB incidence in Viet Nam is particularly high in urban areas.[19] Despite the provision of free diagnostics and anti-TB treatment by the National TB Programme (NTP), high coverage of social health insurance (SHI) covering medical costs[20] and subsidies from the Global Fund for people with DR-TB,[19 21] a 2016 patient cost survey found that 63% of households containing a person with drug-susceptible (DS-TB) and 98% of households containing a person with MDR-TB experienced catastrophic costs due to the disease.[22] The government of Viet Nam is aiming to achieve 100% coverage of SHI by 2025[20] and has announced that in mid-2022, TB care at district-level facilities will be included under the national SHI system.[23] While the full implications of this change are unclear, this likely will increase user fees and out-of-pocket payments for the majority of individuals undergoing DS-TB care, further highlighting the importance of social protection measures for people with TB.

Viet Nam's NTP has identified the impoverishing effects of TB as a priority for action[24] and has established a national fund (Patients Support to Fight TB) to support poor and uninsured people with TB (eg, by purchasing SHI on their behalf).[22 24] However, there is limited context-specific understanding of the optimal ways in which supportive interventions can improve treatment outcomes and reduce costs for people with TB in urban Viet Nam. A prior study exploring the acceptability of cash transfers and SHI among people with TB, healthcare providers (HCPs), policymakers and members of the public in Viet Nam[25] indicated that further research into preferences for supportive interventions was required. Therefore, this study aims to increase understanding of what people with TB and HCPs in urban Viet Nam perceive as important in the design of interventions to reduce costs and improve TB treatment outcomes.

## METHODS

### Study design

We conducted an exploratory qualitative study using semistructured focus group discussions (FGDs) and key informant interviews (KIIs) to understand the perceptions of people with TB and TB HCPs in urban Viet Nam regarding supportive interventions to improve treatment outcomes and reduce costs. The Consolidated Criteria for Reporting Qualitative Research checklist[26] was used to ensure comprehensive reporting.

### Study setting

The study was conducted in two cities in Viet Nam, Ha Noi and Ho Chi Minh City, where a consortium of local non-governmental organisation (NGO) partners works with the NTP to deliver enhanced TB care and conduct operational research.[27 28] Data were collected in July–September 2020 and February 2022 in the context of COVID-19.

At the time of the study, the NTP provided anti-TB drugs and consultations free-of-charge in the public sector. SHI could not be used to subsidise first-line TB medications[22] at district-level facilities but covered 80% of eligible medical expenditure for the majority of people.[21] To obtain SHI, people must register and pay an annual fee, typically for their entire household.[29] For most people, SHI is registered at the commune or district level and is not valid at provincial or national hospitals without a referral.[30] People with TB meet other costs of treatment through out-of-pocket payments and social protection schemes supported by government programmes, the Global Fund and other funding sources.[22]

In 2021, the TB incidence rate in Viet Nam was 173 cases per 100 000 people and there were over 2600 new cases of MDR-TB.[1] Among those treated, treatment was successful in 91% of new and relapse cases and in 74% of MDR-TB cases.[1]

### Participant selection

Purposive sampling was used to select participants with a range of experiences of TB treatment who had not previously engaged in the NGOs' services. TTN contacted District TB Units (DTUs) in Ha Noi and Ho Chi Minh City who provided lists of potential participants: people who had been successfully or unsuccessfully (unsuccessfully treated defined as retreated or relapse cases) treated for bacteriologically-confirmed DS-TB, people being treated for bacteriologically-confirmed MDR-TB with a 9 or 20 month regimen and TB HCPs employed at national, provincial and district levels.

An assessment of information power[31] concluded that a moderate sample size was required to address the study's aim. The sample size was guided by estimates based on previous studies and data saturation was discussed

**Table 1** Participant demographic characteristics

| | Sex (n) | | |
| | Female | Male | Total (n) |
| --- | --- | --- | --- |
| All participants | 12 | 21 | 33 |
| Location | | | |
| Ha Noi | 10 | 12 | 22* |
| Ho Chi Minh City | 2 | 9 | 11 |
| Participant group | | | |
| People successfully treated for DS-TB | 1 | 7 | 8 |
| People unsuccessfully treated for DS-TB | 0 | 6 | 6 |
| People with MDR-TB | 4 | 6 | 10 |
| District TB healthcare providers (HCPs) | 3 | 2 | 5 |
| National/provincial TB HCPs | 3 | 1 | 4 |

*At least one participant was accompanied by their caregiver. DS-TB, drug-susceptible TB; MDR-TB, multidrug-resistant TB; TB, tuberculosis.

following each FGD or interview. A total of 66 individuals were invited, but 33 (13 men, 20 women) declined to participate, mostly due to a lack of interest or distance from the interview site. One participant was accompanied by their caregiver, who also gave consent to participate. Demographic characteristics are reported in table 1.

### Data collection
Data were collected by female NGO staff trained and experienced in conducting qualitative research, and data collection was directly overseen by TTN, a qualitative researcher and NGO project manager with 5 years of qualitative research experience. All participants were informed about the purpose of the study when they were invited to participate by telephone, although they were not provided with details about the interviewers' motivations or biases. They provided written informed consent prior to data collection. Participants were provided with refreshments and 300 000 VND (US$12.73, 23 571.8 VND: US$1, xe.com) to cover travel costs.

Four FGDs with people with DS-TB and one FGD with DTU HCPs were conducted in July–September 2020 in dedicated offices. Two FGDs with people with MDR-TB and four interviews with national/provincial TB HCPs were conducted in February 2022 at the Ha Noi Lung Hospital, the NGO offices or over Zoom. FGDs were conducted in person and lasted 91–188 min. KIIs were conducted via Zoom and lasted 46–90 min. The study aimed to recruit five participants per FGD, however some were smaller due to enrolment challenges and may be better classified as group interviews. No repeat interviews were conducted. Further details can be found in online suplemental appendix A.

Topic guides were developed iteratively by TTN, RF and KS and included questions on challenges during TB treatment and different forms of support; complete topic guides are given in online supplemental appendix B. Questions on digital support were added to later FGDs and interviews due to its increased use during the COVID-19 pandemic. Participants were also asked to rank interventions that they thought were most important for improving treatment outcomes and reducing costs.

FGDs were conducted in Vietnamese, audio recordings were transcribed by NGO staff and then translated into English by an external service provider in Viet Nam. The translations were checked by bilingual study staff. Field notes were recorded following the FGDs and KIIs. Participant data were pseudonymised, stored securely and not shared beyond the study team. Transcripts were not shared with participants for comment.

### Patient and public involvement
People with TB, HCPs, policymakers and members of the public were involved in this research through the development of topic guides. Their responses to a prior qualitative interview study on the acceptability of cash and SHI[25] informed the development of a list of forms of social protection to explore further in this study. The results of this research were disseminated to government officials, the NTP and patient communities in roundtable discussions.

### Data analysis
Data were analysed using reflexive thematic analysis.[32 33] IS, a global health master's student, developed the coding framework with SA, an experienced qualitative researcher, and coded the data using 'Dedoose' software (V.9.0.54)[34] before identifying themes based on recurrent patterns. The analysis was conducted through an iterative process. Codes, categories, themes and subthemes were verified by SA, KS and RF. An example is illustrated in table 2.

Themes were interpreted in line with the Framework for Transformative Social Protection which defines social protection as a range of policies or initiatives which help vulnerable people to move sustainably out of poverty by acting on their financial risk and enhancing their social status.[35]

### RESULTS
We generated three themes and five subthemes from the data, summarised in table 3 and discussed below. Participants also spoke about challenges experienced during TB treatment which are covered extensively elsewhere. We have indicated where HCPs and people with TB differed in opinion, and used 'participants' where they concurred.

**Table 2** Examples of quotes with corresponding code descriptions, codes, themes and subthemes

| Example quote | Code description | Code | Subtheme | Theme |
|---|---|---|---|---|
| There are many aspects that make people give up including travel expenses. So, thanks to this voucher support during the treatment a lot of sick people would benefit. | The costs of TB are a barrier to completing treatment | Financial problems affect adherence | Enabling people with TB to meet medical costs is a priority for support | Existing financial safety nets are essential, but could go further to support people with TB |
| When I was sad, my family encouraged me. | People with TB are encouraged and supported by friends and family | Emotional support from friends and family | Psychosocial support is fundamental for people with TB to 'feel secure in treatment' | It is important to promote 'physical and spiritual health' during TB treatment |
| Besides, during the time of taking this medicine, it was very tiring and exhausting, so I had to drink a lot of water, take tonics during that course. | TB drugs can make people with TB feel tired and weak | Treatment makes you tired and weak | Nutritional supplementation is viewed as important for people with TB to be able to 'fight against the disease' | It is important to promote 'physical and spiritual health' during TB treatment |
| When they know it, honestly, they are afraid to examine and test. If they are ignored about how it is dangerous, they shall still omit it. | Contacts of people with TB do not want to get tested | Contacts reluctant to test | Increasing access to TB diagnostic and preventive services is an important part of social protection | Accessibility and acceptability are critical in designing social support interventions for people with TB |

TB, tuberculosis.

## Theme 1: existing financial safety nets are essential but could go further to support people with TB

### Subtheme 1.1: enabling people with TB to meet medical costs is a priority for support

Participants were concerned about the financial burden of TB and viewed financial support as essential to reducing costs and ensuring good treatment outcomes. Most participants, particularly HCPs, thought that people with TB should not pay any medical costs during their treatment and noted this as their top priority for reducing costs and improving treatment outcomes. Participants valued provision of treatment by the NTP, but many people with TB reported paying different amounts depending on treatment location, treatment regimen or insurance status.

Participants described SHI as important, but difficult to access. Many people with TB reported that SHI registration was slow and expensive (particularly for households) and many were not registered at the right level. In fact, people with TB and HCPs alike reported that seeking care outside of the primary registered health facility was challenging. HCPs also noted difficulties with making claims, calculating payments and deductibles.

However, many participants prioritised SHI for reducing the costs of TB care. One male DTU HCP summarised

**Table 3** Themes and subthemes identified during reflexive thematic analysis

| Themes | | Subthemes | |
|---|---|---|---|
| 1 | Existing financial safety nets are essential but could go further to support people with TB | 1.1 | Enabling people with TB to meet medical costs is a priority for support |
| | | 1.2 | Flexible financial support can give people with TB more agency |
| 2 | It is important to promote 'physical and spiritual health' during TB treatment | 2.1 | Psychosocial support is fundamental for people with TB to 'feel secure in treatment' |
| | | 2.2 | Nutritional supplementation is important for people with TB to be able to 'fight against the disease' |
| 3 | Accessibility and acceptability are critical in designing social support interventions for people with TB | 3.1 | Support for people with TB can be prioritised by need—there is no one-size-fits-all solution |
| | | 3.2 | Support should alleviate the burden of people with TB, not add to it |
| | | 3.3 | Increasing access to TB diagnostic and preventive services is an important part of social protection |

TB, tuberculosis.

this view, saying 'We should focus on basic support. For example, health insurance is necessary, then testing'. Overall, participants thought that uninsured people with TB should be provided with SHI, that SHI coverage or the NTP's provision should be expanded to cover more costs including diagnostic and confirmatory testing (including X-rays and GeneXpert), hospitalisation fees and supplements and that registration should be made simpler.

### Subtheme 1.2: flexible financial support can give people with TB more agency

Many participants thought that cash transfers were important for mitigating out-of-pocket expenses and income loss and improving treatment outcomes. For people with TB in particular, cash transfers were a priority because they saw them as simple, quick and versatile and could be used to solve upfront costs such as hospitalisation fees. One male participant who was successfully treated for DS-TB expressed that 'Cash, in general. Cash solves many things. It's simple'. Further, cash transfers were seen by most as easier to implement than vouchers, easier to access than loans and could give people with TB more independence to spend at their discretion. However, other people with TB spoke about a potential to misuse cash transfers (eg, purchasing tobacco) and therefore thought that transport or nutrition vouchers were a better way to provide financial support.

People with TB also wanted support to find suitable employment to counter lost income. They indicated that career counselling, help with finding a new, less physically demanding job through connection with a job centre or job placement, or lending money to invest in their career would be beneficial. This was seen as more useful than vocational support, with one male participant who was successfully treated for DS-TB saying 'After training, there must be available jobs, or else, it's just a waste of time, it's very difficult to re-train'.

### Theme 2: it is important to promote 'physical and spiritual health' during TB treatment

### Subtheme 2.1: psychosocial support is key for people with TB to 'feel secure in treatment'

People with TB thought that financial support could also reduce worry and provide a sense of security during treatment, and thus improve mental health. This was summarised by one participant who was unsuccessfully treated for DS-TB who said that 'Sometimes, it is only a small part of a support package, but it provides them (the beneficiary) with something for their spirit- to help them make their best effort in their treatment. They feel better and cared for by society'.

All participants spoke about the negative impact that TB has on mental health. One female national/provincial TB HCP described it as: 'There are patients who know they are depressed, bored, feel that they are not getting better, they are still tired from taking medication, the illness, they want to give up'. More broadly, people with TB frequently referred to how pessimism can make

it harder to recover and about the importance of feeling secure in their treatment.

As a result, many participants recognised a need for psychological, emotional and spiritual support to promote better treatment outcomes. People with TB described emotional support from friends and relatives as essential, but recognised a need for increased access to a psychologist or psychiatrist, particularly for people with MDR-TB. Similarly, TB HCPs identified a need for psychological training given the scarcity of mental health specialists, as highlighted by this female national/provincial TB HCP 'Psychiatrists are very, very rare. Even our [national-level] hospital does not have any. So sometimes, we see our patients and sincerely share from our hearts and our thoughts. But in terms of psychological-expertise, we have no specialisation in psychology'. However, some people with DS-TB reported no psychological problems and were reluctant to engage with proposed support.

Participants also thought that patient support groups could help people with TB to learn from one another and encourage optimism about treatment. This was summarised by one female participant with MDR-TB as 'When we undergo the treatment, we talk to people who don't have the disease, they won't understand us… But it's easier to talk to people who feel the same way as we do'. Participants thought that these groups should be led by experts, whether that be HCPs, mental health specialists, or others with TB.

For many participants, particularly people with MDR-TB, comprehensive TB education and guidance from HCPs about TB treatment was an important part of feeling secure in their treatment, for example by helping them to manage side effects. People with TB thought that information should be provided by doctors at the start of treatment and available throughout. One participant with MDR-TB suggested that specialist pharmacists would be able to answer questions about side-effects of medicines. HCPs also suggested using leaflets, recorded messages and public media to share information about TB prevention and treatment.

### Subtheme 2.2: nutritional supplementation is viewed as important for people with TB to be able to 'fight against the disease'

Most participants thought that good nutrition and rest were essential to recovery and a priority for support, particularly for people with MDR-TB. One HCP at a DTU summarised that 'tuberculosis is exhausting. If you are malnourished, you need energy, fibre, and vitamins'. Some participants thought that people with TB should be supported directly with food but many considered cash to be more helpful.

Participants also thought that supplements and tonics were essential for controlling side effects (eg, through detoxifying the liver). All participants thought that they should be included in a support package rather than paid out-of-pocket. One national/provincial TB HCP highlighted how costs can accumulate throughout treatment by explaining that 'Some patients don't have money to

buy them, so they just receive the TB medications, no supplements. Then they have liver and kidney issues and get hospitalised. Then, the cost of liver and kidney detox would be many times higher than the cost of liver supplements they should have paid for in the beginning'.

### Theme 3: accessibility and acceptability are critical in designing social support interventions for people with TB

#### Subtheme 3.1: support for people with TB can be prioritised by need—there is no one-size-fits-all solution

Participants were asked who should be prioritised to receive each type of support and when it should be delivered. Some participants in each group thought that all people with TB should be supported equally, reasoning that prioritisation would cause jealousy, and that it should last throughout treatment. However, most participants thought that support for people with TB should be tailored to their individual needs.

Participants thought that people struggling financially or with more severe disease had greater needs for support and should therefore be prioritised. For example, one male participant successfully treated for DS-TB suggested that 'TB patients should be all supported, dividing by severity or circumstances, the poor or persons living in difficult situations, second treatment or drug resistance'. Similarly, when prompted to prioritise when support was given, many people with TB suggested it be given during the intensive phase of treatment. On the other hand, many national/provincial TB HCPs were explicit that support should be highly specific, with transport vouchers according to distance travelled or financial support according to days of hospitalisation.

#### Subtheme 3.2: support should alleviate the burden of people with TB, not add to it

Speaking from past experiences, TB HCPs raised concerns that defining eligibility criteria for support would create overly burdensome registration systems that impede access. One district TB HCP summarised that 'If you offer, but in the end you can't support due to the inflexible procedures… You support nobody.'

People with TB were also concerned with the practicality of proposed support options. Transport vouchers were viewed positively because they help people attend appointments and adhere to treatment. However, delivering food was viewed as a waste of time and resources. Similarly, people with TB viewed vocational training as impractical with one male participant who was successfully treated for DS-TB saying 'As old people have a certain job, it's hard to change to another. It's okay to just introduce an easier job for them, vocational training is not feasible.'

Additionally, participants wanted support to be convenient and accessible. They viewed current support as insufficient or inaccessible due to inflexible procedures or inadequate publicity. People with DS-TB viewed patient support groups positively but were reluctant to participate due to the time commitment and fears over re-infection.

District TB HCPs had concerns about adding to their own overburdened workload by running these groups in person. Participants thought that online support groups would increase access to support in the context of limited staff resources and suggested using online forums on social media platforms to ask questions and share information. However, they had concerns about the accessibility of online support for older people.

Similarly, people with MDR-TB and national/provincial TB HCPs viewed remote delivery of medication and online consultations as an additional way to access support when necessary, such as during periods of intense COVID-19 restrictions but people with TB did not want to replace in-person appointments entirely.

#### Subtheme 3.3: increasing access to TB diagnostic and preventive services is an important part of social protection

Many participants thought that any support given should extend beyond the individual. Access to testing and preventive treatment for families affected by TB was prioritised by many people with TB and national/provincial TB HCPs as an important way to reduce onwards transmission and reduce costs. People with TB suggested that their families be provided with home testing kits or transport vouchers to enable testing without incurring costs. One individual with MDR-TB highlighted the potential socioprotective effects of doing so, saying 'It's good for my family to go to the doctor to get screened… now there's a support system, it'll help with the expenses… If they go for screening, the disease is detected early, good for them. If you don't do it until it's severe, treatment will be more difficult.'

## DISCUSSION

This study demonstrates that people with DS-TB and MDR-TB and TB HCPs perceive mixed interventions that include social support and social protection measures to be important for reducing costs and improving TB treatment outcomes. In particular, measures that 'matter most' include provision of SHI and flexible cash transfers to meet the costs of treatment, nutritional support that enables people with TB to manage their side-effects, and psychosocial support that encourages them during treatment. Recommendations for the design of supportive interventions, as informed by our findings regarding participants' priorities, are summarised in table 4.

To the best of our knowledge, this is the first study to explore how people directly affected by TB prioritise a broad range of different social protection and psychosocial support interventions. This study adds to existing evidence by showing that both people with TB and TB HCPs prioritise interventions that extend beyond SHI and cash transfers, which are currently available in urban Viet Nam and are most common in social protection interventions globally.[16] Furthermore, it highlights the broader benefits of social protection, demonstrating how the provision of any support can make people with TB

**Table 4** Recommendations for action based on participant responses

| Theme | Recommendation |
|---|---|
| Existing financial safety nets are essential but could go further to support people with tuberculosis (TB) | ▶ Ensure all people with TB are covered by Viet Nam's National Social Health Insurance scheme, and plug gaps in coverage for those who are not<br>▶ Provide cash transfers to people with TB to be spent at their discretion<br>▶ Establish, or link people with TB to existing, career counselling services or job placements |
| It is important to promote 'physical and spiritual health' during TB treatment | ▶ Increase availability of trained psychologists and psychiatrists where possible, or provide psychological training for TB healthcare practitioners (HCPs), particularly for people with multi-drug resistant TB (MDR-TB)<br>▶ Establish online peer-support forums using platforms such as messenger applications (eg, WhatsApp or Zalo) involving people with TB and TB HCPs to act as an information-sharing platform and enable patients to report and receive assistance for side-effects<br>▶ Ensure comprehensive TB education to help people with TB manage their treatment and reduce worry, including guided counselling sessions between people with TB and TB doctors at the beginning of treatment as standard of care |
| Accessibility and acceptability are critical in designing social support interventions for people with TB | ▶ Continue and extend existing transport and nutritional support, particularly for people with MDR-TB<br>▶ Increase access to testing and treatment for TB infection and ensure all contacts of people with TB are tested for TB<br>▶ Differentiate all support according to multidimensional need, considering economic status and severity of disease as appropriate |

feel more secure in their treatment, as well as mitigating its costs.

These findings can be considered in keeping with a framework for transformative social protection (figure 1),[35] which extends the definition of social protection beyond financial risk protection to include a broader collection of promotive, protective and transformative

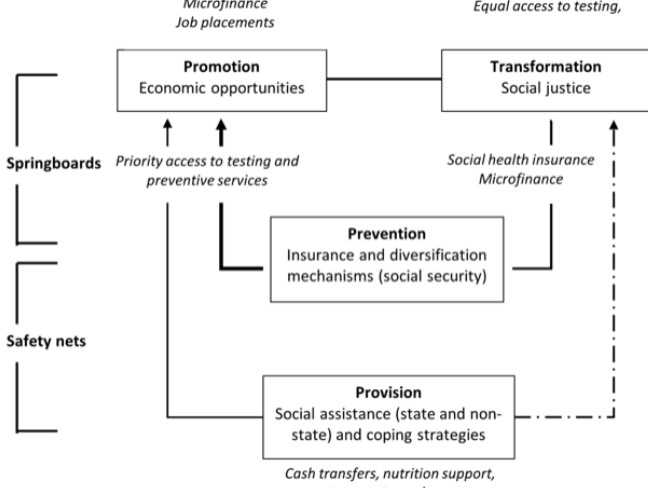

**Figure 1** A conceptual framework for social protection, modified from Sabates-Wheeler and Devereux.[30] Thicker lines indicate stronger relationships. The dotted line indicates where measures could have counterproductive effects on social transformation. The top half of the diagram represents policies that can facilitate movement out of poverty (springboards), while the bottom half represents those that alleviate and prevent deprivation (safety nets). Examples relevant to TB in italics. TB, tuberculosis.

interventions that provide both 'safety nets' and 'springboards' out of poverty.

Universal health coverage, including access to health insurance, is a key part of the WHO's End TB Strategy. In this study, many HCPs prioritised SHI over other interventions and suggested that SHI, or existing subsidies, should be extended to cover all treatment costs for all people with TB. Participants also recognised challenges with its implementation, reflecting recent findings in this setting[36] and that catastrophic costs remain high despite high SHI coverage due to non-medical costs and lost income, in line with previous studies.[2 22] Therefore, and as recognised by the NTP,[24] supportive interventions for people with TB in urban Viet Nam should extend beyond SHI.

There is mounting evidence for the positive impact of cash transfers on reducing costs and improving treatment outcomes for people with TB.[14 15 37] Therefore, it is perhaps unsurprising that all participant groups in this study identified cash transfers as a priority for both improving outcomes and reducing costs. Furthermore, our study highlighted that cash transfers give people with TB more agency and are simple to manage, as well as being more cost-effective.[38] Cash transfers are already provided for poor or uninsured people with TB by Viet Nam's government and for people with DS-TB in certain regions of Ha Noi and Ho Chi Minh City under donor projects,[39] so expanding to more people with TB may bring wider benefits.

However, there is an ongoing debate regarding whether cash, vouchers or in-kind support are more effective.[38] Targeted provision measures, such as transport or nutrition vouchers, can provide a 'safety net' to avoid the

exacerbation of catastrophic costs. Our study demonstrates that alleviating transport costs is also important for people with TB both for improving treatment outcomes, complementing previous findings that transport vouchers can improve treatment adherence and success rates.[40 41] Participants also viewed nutritional and supplement support as key to managing side effects and associated costs, reiterating previous findings.[42] However, the evidence for the clinical benefit of nutritional supplementation[43] or of traditional supplements[44] is limited. Consequently, cash transfers and transport vouchers may be a better way to address non-medical costs, with further research required around the benefits of nutritional and traditional supplements.

Promotive interventions that support people with TB to establish a sustainable livelihood can provide a 'springboard' out of poverty[3 35] by targeting the high burden of lost productivity during TB treatment,[6 22 45] as well as contributing to improved TB treatment outcomes.[3] In this study, we found that job placements or career counselling were preferred over vocational training and could be an effective way to support people with TB, particularly breadwinners and younger people, to re-enter the workforce. However, there must be opportunities for employment available after such interventions. Given the adverse impact of COVID-19 on employment in urban Viet Nam,[46] enabling employment for people with TB may be more important than ever.

Social protection is one part of a broader set of efforts needed to contain TB.[8] Participants in our study also emphasised the importance of social support. In particular, they noted the importance of increasing access to psychological support given the distress experienced during TB treatment.[17 47 48] Integrating psychological support into existing care by training TB HCPs, as suggested by our participants, could improve access and has been shown to be effective in another context.[49] Informal online patient support groups could also promote wider access to psychosocial support and TB education by allowing people with TB to learn from one another. These may become more acceptable following the COVID-19 pandemic, noting potential difficulties for older people. In line with the framework for transformative social protection, which emphasises both financial and social aspects of vulnerability,[35] there is evidence that combined psychological and financial interventions are more effective at improving treatment outcomes than financial support alone.[17 50] This could therefore improve supportive interventions for people with TB in urban Viet Nam.

Our participants highlighted the importance of accessibility of TB services and support both for people with TB and their families. This is in line with a recent scoping review which identified a lack of awareness and logistical complexities as key barriers to improving equity through social protection.[16] One way in which this could be achieved is through active case finding (ACF) to identify unidentified persons with TB. This is especially important in Viet Nam given that just 46% of estimated new TB cases in 2021 were notified to the government.[1] Viet Nam's NTP has recognised the importance of ACF as it can reduce transmission and improve health outcomes over time. Furthermore, there is evidence from Ho Chi Minh City that it can reduce the incurrence of catastrophic costs and can reach more marginalised individuals, increasing equity in TB care.[51] ACF could therefore be considered a key part of a wider package of social protection measures in this context and contribute to achieving the End TB goals.[7]

Finally, our study suggests that supportive interventions for people with TB could be prioritised to make effective use of resources by allocating based on individual need, taking the agency of people with TB in identifying these needs into account and targeting the intensive phase of treatment when costs are highest. Differentiated service delivery, in which services are tailored to individual needs, is already a key aspect of person-centred HIV care that has been shown to improve access to and uptake of testing and treatment.[52] A similar approach could be taken to allocating packages of support for people with TB, learning from experiences with HIV to ensure that any eligibility criteria did not create barriers to accessing support. However, modes of differentiating support are highly context dependent[40 52] and further research is required to identify optimal approaches for people with TB in Viet Nam.

### Strengths and limitations

A key strength of this study is the inclusion of a diverse group of participants, enhancing trustworthiness and validity and ensuring that the resulting recommendations take their views into account and can therefore inform person-centred and context appropriate intervention design. Some argue that conformity pressures can lead to 'groupthink' in FGDs.[53] However, participants expressed some different opinions on the best way to support people with TB with the costs of treatment and there was broad consensus between participants in different FGDs, suggesting robustness of the findings. Interviews were conducted in the NGO offices rather than in the health facility to mitigate social desirability bias.

A limitation is that this study was conducted solely in urban settings in Viet Nam, therefore the findings may not be generalisable to rural settings within Viet Nam or to other high-burden countries. The needs of people with TB may have shifted since data collection given the widespread impact of COVID-19 on health service utilisation and economic well-being[54 55] as well as changes to financing for TB.[23]

Further, the analysis was conducted on English translations of the transcripts rather than in Vietnamese, the original language of data-collection, which may have influenced their interpretation.[56] However, the study team took a rigorous approach to transcription and translation and there was close collaboration between

Vietnamese-speaking and non-Vietnamese-speaking team members to limit the effect of ethnocentrism on the analysis.

## IMPLICATIONS AND CONCLUSIONS

Our study highlights that a broad range of social protection and social support measures are required to meet the needs of people with TB in urban Viet Nam and identifies interventions that are important to people with TB and HCPs in this context. While Viet Nam has begun to implement social support for people with TB, we found that interventions that extend beyond the health system are needed to better protect people with TB against costs and to improve treatment outcomes.

In conjunction with prior research, this study can play an important role in the design of supportive interventions for people with DS-TB and MDR-TB. Within urban Viet Nam, it can guide how government funds, such as the Patient Support to Fight TB (PASTB) fund, support people with TB with recommendations for actions outlined in table 4. To expand on this, future research could explore the optimal operational design of interventions, for example by evaluating whether online support groups or psychological training for HCPs are the best way to support people with TB with their mental health.

More broadly, this study emphasises that coordinated action from biomedical, social and economic sectors is needed to support people with TB and other diseases of poverty in Viet Nam and other high-burden countries.

**Author affiliations**
[1]Department of Global Public Health Sciences, WHO Collaborating Centre for Social Medicine and Tuberculosis, Karolinska Institutet, Stockholm, Sweden
[2]Friends for International TB Relief, Ha Noi, Viet Nam
[3]Centre for Development of Community Health Initiatives, Ha Noi, Viet Nam
[4]IRD VN, Ho Chi Minh City, Viet Nam
[5]Stop TB Partnership, Geneva, Switzerland
[6]USAID Vietnam, Ha Noi, Viet Nam
[7]National TB Program, National Lung Hospital, Ha Noi, Viet Nam
[8]Faculty of Social Sciences, Tampere University, Tampere, Finland

**Acknowledgements** Our special thanks go to the participants in this study for their invaluable contributions to this study. We are particularly grateful to the people with tuberculosis (TB) for sharing their personal experiences of illness and hope that this study will have a positive impact on people with TB in the future. We would like to thank the individuals who conducted interviews including: Nham Thi Yen Ngoc, Tran Thi Ngan and Bui Thi Huyen. Special thanks to the individuals who completed transcriptions, translations and the checking of these processes, including Nguyen Thi Cam Van, Truong Thi Thuy Dung, Nguyen Thi Xuan, Phan Khanh Hang and Huynh Thi Kim Ngan. We appreciate the staff of the National Lung Hospital, Ha Noi Lung Hospital and Pham Ngoc Thach Hospital who facilitated interviews for our research team. TB staff based in the following districts of Ha Noi assisted with participant recruitment: Ba Dinh, Dong Da, Hai Ba Trung, Hoan Kiem and Hoang Mai. In Ho Chi Minh City, we would like to thank staff from five districts who assisted with recruitment including: Binh Chanh, Go Vap, Thu Duc and Districts 6 and 8.This study was made possible by the generous support of the American people through USAID. The contents are the responsibility of the listed authors, and do not necessarily reflect the views of USAID or the US Government.

**Contributors** IS codeveloped the research questions, led data analysis (developing the coding framework and identifying themes), drafted and revised the manuscript. RF conceptualised the study, designed topic guides, oversaw data collection,

supported and verified analysis and reviewed the manuscript. She is the guarantor for the paper. KS conceptualised the study, designed topic guides, codeveloped the research questions and supported in analysis. NTTN conceptualised the study, oversaw data collection, advised on and reviewed the manuscript. LN, THYP, HN, AC, LNQV, NTTN, AK, JC, MPH, LB, KL, HBN and NVN edited and reviewed the manuscript. SA codeveloped the research questions and the coding framework, supported and verified analysis and reviewed the manuscript. All coauthors reviewed and approved the final manuscript.

**Funding** This study was supported by two funders. Interviews, transcriptions and translations among people with DS-TB were funded by the United States Agency for International Development (USAID) Fixed Award No. 72044020FA00001, through the Erase TB Activity. DR-TB interviews were conducted with support from the Stop TB Partnership's TB REACH initiative by Global Affairs Canada grant number CA-3-D000920001.https://w05.international.gc.ca/projectbrowserbanqueprojets/projectprojet/details/d000920001.

**Competing interests** None declared.

**Patient and public involvement** Patients and/or the public were involved in the design, or conduct, or reporting or dissemination plans of this research. Refer to the Methods section for further details.

**Patient consent for publication** Consent obtained directly from patient(s).

**Ethics approval** This study was conducted in strict adherence with the Declaration of Helsinki. Ethical approvals were obtained from Ha Noi University School of Public Health (issued on 17 July 2020; IRB #300/2020/YTCC-HD3) and from Pham Ngoc Thach Hospital (issued on 16 September 2020; IRB # 1225/PNT-HĐĐĐ). Participants gave informed consent to participate in the study before taking part. This study was supported by two funders. Interviews, transcriptions and translations among people with DS-TB were funded by the United States Agency for International Development (USAID) Fixed Award No. 72044020FA00001, through the Erase TB Activity. DR-TB interviews were conducted with support from the Stop TB Partnership's TB REACH initiative by Global Affairs Canada grant number CA-3-D000920001 https://w05.international.gc.ca/projectbrowserbanqueprojets/projectprojet/details/d000920001.

**Provenance and peer review** Not commissioned; externally peer reviewed.

**Data availability statement** Data are available upon reasonable request. A datasharing agreement was reached between Friends for International TB Relief and the Karolinska Institutet. FIT collected data directly or obtained it through collaboration agreements with local collaborating NGO partners. All data were pseudonymised prior to sharing and stored on a secure, password-protected online platform at Karolinska Institutet.

**ORCID iDs**
Rachel Forse http://orcid.org/0000-0002-0716-3342
Luan Nguyen Quang Vo http://orcid.org/0000-0002-5937-6286
Binh Hoa Nguyen http://orcid.org/0000-0002-1543-4907

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
