## [Reviewer comments · BMJ Open]

ARTICLE DETAILS

TITLE (PROVISIONAL)	What matters most? A qualitative study exploring priorities for supportive interventions for people with tuberculosis in urban Viet Nam
AUTHORS	Smith, Isabel; Forse, Rachel; Sidney, Kristi; Thanh, Nguyễn Thị; Nguyen, Lan; Phan, Thi Hoang Yen; Nguyen, Han; Codlin, Andrew; Vo, Luan; Nguyen, Nga Thi Thuy; Khan, Amara; Creswell, Jacob; Pham Huy, Minh; Basu, Lopa; Lönnroth, Knut; Nguyen, Binh Hoa; Nguyen, Viet Nhung; Atkins, Salla

VERSION 1 – REVIEW

REVIEWER	Hoddinott, Graeme Stellenbosch University Faculty of Medicine and Health Sciences, Desmond Tutu TB Centre
REVIEW RETURNED	01-Jun-2023

GENERAL COMMENTS	Congratulations to the co-authors on completing important work, finding ways to bring patient and provider perspectives into the design of TB services. Overall, the work is impressive. I make two substantive and a few minor suggested revisions. Substantive (suggested) revisions 1. Overall, the balance between the premise of the paper (as outlined in the background) and the discussion seems off. Much of what is presented in the discussion could also have been part of the background. Further, the discussion does not adequately highlight what is novel in the findings and what is a confirmation / similar to what we knew before this paper from other studies. This is an issue of organization / punchiness of the writing. The first paragraph of the discussion should summarize the finding highlights and the next paragraph directly answer what is new about these in overview. Then the discussion overall could be shortened. 2. Relatedly, I suggest that table 4 should be the primary organization of the findings. Whilst the themes and sub-themes may have been useful analytically, the stated objective is more about what people said we should do to improve SHI delivery (table 4) than it is about a way of grouping the content of everything reported in the discussions. The themes and sub-themes at times are equivocal in what the key messages are, 'some people say X, others say Y, etc.'. Instead, the findings could be organized by the three sub-sections presented in table 4 with data drawn from whichever discussion to show what is meant and who agreed / disagreed with this recommendation. Minor (discretionary) revisions Page 7, line 12 - missing word: "and used 'participants' they
--

	concurrent'. (perhaps 'where' the concurrent?). Page 14, line 16 - missing space between '(ACF)to ...' Sub-theme 3.1 - what about the patient's agency in selecting the support they need most? Page 14, lines 41-47 - On the other hand, these could just be the 'lowest common denominator' set of opinions? Page 14, lines 56-60 - I'm not sure that this needs to be mentioned specifically as a limitation. Most high burden TB settings are places where English is not the primary language. And many of the study team and co-authors are presumably Vietnamese fluent. Conversely data collected in the participants' first language are often far richer as they are better able to express themselves. If we start thinking that collecting data in local language is a limitation then I don't know any more. Page 15, lines 13-21 - in every sentence the authors write X 'could' Y. Since this is a conclusion, can we be more definite about what 'should' be done? Page 15, lines 23-32 - Unnecessary? Part of the premise of the study rather than a conclusion.
--	--

REVIEWER	Mbuthia, Grace Jomo Kenyatta University of Agriculture and Technology
REVIEW RETURNED	04-Jul-2023

GENERAL COMMENTS	The authors have studied an important area and the study findings are crucial in informing Tuberculosis control programs on areas of improvement for better Tuberculosis treatment outcomes not only in urban Viet Nam but also countries with similar settings. The choice of exploratory qualitative design involving both patients and health care provider was well suited for the research questions in the study. The article is well written however in the abstract the conclusion could be improved to specifically answer the question what matters most ? .
---

VERSION 1 – AUTHOR RESPONSE

Reviewer: 1	
Dr. Graeme Hoddinott, Stellenbosch University Faculty of Medicine and Health Sciences	
Substantive (suggested) revisions	Response
1. Overall, the balance between the premise of the paper (as outlined in the background) and the discussion seems off. Much of what is presented in the discussion could also have been part of the background. Further, the discussion does not adequately highlight what is novel in the findings and what is a confirmation / similar to what we knew before this paper from other studies. This is an issue of organization / punchiness of the writing. The first paragraph of the discussion should summarize the finding highlights and the next paragraph directly answer what is new about	We agree and have amended the discussion in line with your suggestion. Specifically, to address this we have:  • Edited the first paragraph to more clearly summarise the central findings of the study • Added a second paragraph that underscores what this study adds to previous research • In the following paragraphs that

these in overview. Then the discussion overall could be shortened.	explore the findings further, we have moved some content to the introduction where there was repetition or where it made more sense (e.g. introducing different forms of social protection and evidence for their use in TB).
2. Relatedly, I suggest that table 4 should be the primary organization of the findings. Whilst the themes and sub-themes may have been useful analytically, the stated objective is more about what people said we should do to improve SHI delivery (table 4) than it is about a way of grouping the content of everything reported in the discussions. The themes and sub-themes at times are equivocal in what the key messages are, 'some people say X, others say Y, etc.'. Instead, the findings could be organized by the three sub-sections presented in table 4 with data drawn from whichever discussion to show what is meant and who agreed / disagreed with this recommendation.	We have not re-organised the results section as we believe the themes and sub-themes from the thematic analysis are important to include in the interest of transparency, as they are closer to the original data and show how we have arrived at our recommendations. We have also chosen not to draw out individual cases of who agreed and did not, aside from where there was a clear consensus, as the analysis was conducted at the group rather than individual level. Instead, to address this comment we have re-organised the table so that the recommendations are in the same order as the relevant parts of the discussion, and brought it up to the top of the discussion with the summary of our findings such that it answers the question of 'what matters most?' more directly. We have also restructured the table to make the recommendations clearer to read.
Minor (discretionary) revisions	Responses
Page 7, line 12 - missing word: "and used 'participants' they concurred'. (perhaps 'where' the concurred?).	Error amended.
Page 14, line 16 - missing space between '(ACF)to ...'	Error amended.
Sub-theme 3.1 - what about the patient's agency in selecting the support they need most?	We agree this is a key point, but as it was not raised in the FGD we feel it is better placed in the discussion section. We have added a sentence on page 14 to this effect.
Page 14, lines 41-47 - On the other hand, these could just be the 'lowest common denominator' set of opinions?	We interpreted this as a comment about conformity pressures in FGDs, and have added in this counter-point to our discussion of strengths and limitations.
Page 14, lines 56-60 - I'm not sure that this needs to be mentioned specifically as a limitation. Most high burden TB settings are places where English is not the primary language. And many of the study team and co-authors are presumably Vietnamese fluent. Conversely data collected in the participants' first language are often far richer as they are better able to express themselves. If we start thinking that collecting data in local language is a limitation then I don't know any more.	We believe this due to a misunderstanding. We agree that collecting data in the participants' first language (Vietnamese) is important and a strength of the study. We had meant to suggest that it would have been better to also analyse the data in Vietnamese, rather than in English, to ensure that nothing was lost in translation. However, we took every effort to ensure that no meaning was lost through collaborating closely with bilingual members of the study team.
Page 15, lines 13-21 - in every sentence the authors write X 'could' Y. Since this is a	We have rephrased with 'can' instead of 'could' or 'should' which we believe is more

conclusion, can we be more definite about what 'should' be done?	definite while maintaining a positive tone.
Page 15, lines 23-32 - Unnecessary? Part of the premise of the study rather than a conclusion	Agreed that this in its entirety is unnecessary. We have cut this down, but felt it was important to end on a reflection of the bigger, global picture as the findings of this study could be used to inform thinking on social protection in other high-burden TB contexts.
Dr. Grace Mbutia, Jomo Kenyatta University of Agriculture and Technology	
Comments	Response
The article is well written however in the abstract the conclusion could be improved to specifically answer the question what matters most?	We have added a sentence to the conclusions section of the abstract to directly answer the question.